# Benefit of Unsaturated Soil Mechanics Approach on the Modeling of Early-Age Behavior of Rammed Earth Building

**DOI:** 10.3390/ma15010362

**Published:** 2022-01-04

**Authors:** Parul Chauhan, Noémie Prime, Olivier Plé

**Affiliations:** Université Savoie Mont Blanc, CNRS, LOCIE, 73000 Chambéry, France; noemie.prime@univ-smb.fr (N.P.); olivier.ple@univ-smb.fr (O.P.)

**Keywords:** rammed earth, unsaturated soil mechanics, FEM simulations, early-age behavior

## Abstract

Rammed earth has the potential to reduce the carbon footprint and limit the energy consumption in the building sector due to its sustainable characteristics. Still, its use is not generalized due to a lack of understanding of the material behavior, notably its sensitivity to water. The coupled hydro-mechanical behavior has been recently studied in the framework of unsaturated soil mechanics, using suction as the parameter to represent the hydric state. This dependency of the mechanical behavior on the hydric state leads to uncertainty of the drying period required to progress in the construction process. Notably, the drying period before building the next floor is unknown. To determine the drying period, thermo-hydro-mechanical coupled finite element method simulations were carried out on a single wall by using the unsaturated soil mechanics approach and safety criterion recommendations from the practical guide for rammed earth construction in France. It was determined that it takes significant time for the construction of additional floor both in ‘summer-like’ and ‘winter-like’ environmental conditions, whereas the walls were far away from the ultimate failure state. Thus the drying periods were overestimated. It was concluded that the safety criterion from the practical guide is very conservative and drying periods can be reduced without significantly compromising the safety factor.

## 1. Introduction

Rammed earth is a construction technique by which dense load-bearing walls can be made with dynamic compaction of moist soil between removable shuttering. It is an ancient technique, but interest in it has been revived in the recent years due to its sustainable characteristics. It has various advantages as a sustainable alternative to traditional construction materials such as concrete or steel. It has lower embodied energy [1] (around 1% of the energy needed for construction with cement-based material [2]). In addition, a building made with rammed earth requires low operational energy due to its thermo and hygro-regulator effects [3,4,5]. It is fully recyclable in its unstabilized state (i.e., without any additional binder). Finally, it possesses sufficient strength for the construction of 1–2 story buildings (compressive strength of 1–2 MPa according to New Mexico code, 2001 [6] and 1.3 MPa according to New Zealand standard NZS:1998 [7]).

In spite of the numerous advantages of rammed earth with regards to sustainability, various limitations hinders its widespread use for construction purposes. One of the biggest limitations is its sensitivity to water, which makes its use challenging to be generalized. Moisture ingress into the wall can lead to a change in consistency from solid to plastic which can further lead to a drop of mechanical strength and rigidity. During the life-span of the building, the ambient atmospheric conditions (relative humidity and temperature) are continuously evolving, and thus, affecting the mechanical performance.

The link between the hydric state and the mechanical characteristics has been recently studied in the framework of unsaturated soil mechanics [8,9]. Suction has been used as a state parameter to study this hydric influence [10,11,12,13]. In addition, there are various constitutive models in the unsaturated soil mechanics domain that can be used to study rammed earth, since rammed earth is basically unsaturated compacted soil.

The dependency of the mechanical behavior on the hydric state leads to a practical issue during rammed earth construction with multiple floors. When a rammed earth wall is made, it is in a wet state (i.e., at the water content required for optimum compaction), which leads to a lower strength. Once the wall is allowed to dry in the ambient conditions, it gains strength due to suction-induced cohesion [8,14]. Thus, it is necessary to determine the drying period required for the walls to gain sufficient strength, in order to build a floor above it. The behavior of the walls since the construction until it is able to resist serviceable loading is termed as early-age behavior in this article.

In addition to the lack of characterization of the hydric influence on mechanics, there is also a lack of technical guidelines or regulations. Although some countries have their own set of guidelines and standards, there are uncertainties in the design methodologies. In France, the practical guide for construction using rammed earth is “Guide de bonnes pratiques (GBP) Pisé, 2018 [15]”. It provides some recommendations for the drying period in different environmental conditions (3–4 months drying in summer-like conditions and not recommended to build in winter-like conditions), but there are no definitive regulations. In addition, the design rules for rammed earth walls uses empirical relations or recommendations which does not take into account the hydro-mechanical coupling or the unsaturated soil mechanics approach.

Thus, in order to demonstrate that this practical issue of drying can be numerically solved from real hydro-mechanical parameters, the model of single wall is simulated, which takes into account the thermo-hydro-mechanical coupled modeling. The simulations are carried out in different environmental conditions, which are corresponding to Bourget du lac in the Savoie region of France. A methodology to determine the necessary drying period is proposed. Finally, a comparison of the results with the recommendations by GBP, Pisé, 2018 is made.

## 2. Materials

In this study, in order to carry out the numerical simulations, the experimental results of a study previously carried out (Chauhan et al. [16]) are used. Here a brief description about the soil that was used for experimental study is presented. The soil used comes from an existing construction site at La Roche Condrieu in the Auvergne-Rhone Alpes region of France. The particle size distribution curve was plotted for this soil using sieve analysis and hydrometer analysis according to French standard NF P 94-057 [17]. The particle size distribution curve in Figure 1 shows that it contains 12% gravel, 30% sand, 51% silt, and 7% clay. Basic characterization tests were performed on the soil. Its index properties are: liquid limit $w_{l}=27.42\%$, plastic limit $w_{p}=16.39\%$, and plasticity index is low, $I_{p}=w_{l}-w_{p}=11.03\%$. It is classified as low plastic silt ($I_{p}<12\%$) according to the French classification GTR (Guide de Terrassements Routier) for fine soils. The activity of clay ($A_{c}=I_{p}/f$) defined as the ratio of plasticity index ($I_{p}$) and percentage of soil passing 2 μm sieve (f) was equal to 1.44. The relatively low value of specific surface area ($S_{sp}=14.7$ m${}^{2}$/g) and Cation exchange capacity (CEC = 2.6 cmol/g) suggests a very low percentage of swelling clays which makes the soil suitable for building construction. The mechanical and hydric characterization has been described fully in Chauhan et al., 2019 [16]. Some of the relevant results are mentioned here for the determination of the material parameters required to perform the coupled simulations.

An optimum water content of 12.5 % and corresponding maximum dry density of 1900 kg/m${}^{3}$ were determined from the standard Proctor test. For the determination of the soil water retention curve, the liquid-vapor equilibrium method using saline solutions and the pressure plate method was used. In the liquid-vapor equilibrium method, the relative humidity of the atmosphere is controlled and imposed around the specimen using saline solutions. The saturated aqueous solution of different salts (Table 1) are used to control the relative humidity of the air around the samples from 9% to 97.3% at 25 °C [19]. Based on the difference of relative humidity between the ambient conditions of the surrounding air and the specimen, water exchanges take place by means of vapor transfer up to the equilibrium of the pore relative humidity. The relationship between the equilibrium of relative humidity of the pore air and the suction imposed on the specimen is based on Kelvin’s thermodynamic equilibrium:(1)$$s=u_{a}-u_{w}=-\frac{\rho_{w}\cdot R\cdot T}{M_{w}}ln\left(RH\right)$$
where, *s* is the suction defined as the difference of pore air pressure ($u_{a}$) and pore water pressure ($u_{w}$) at a given temperature *T* (in Kelvin, K), *R* is the universal gas constant (*R* = 8.3143 J/mol/K), $M_{w}$ is the molar mass of water ($M_{w}=0.018$ kg/mol), $\rho_{w}$ is the bulk density of water ($\rho_{w}=1000$ kg/m${}^{3}$) and RH is the relative humidity, which is defined as the ratio of partial vapor pressure P in the considered atmosphere and the saturation vapor pressure $P_{o}$ at temperature (*T* = 298 K).

Small samples were compacted using a method of static double compaction. In this method, a compaction pressure of 5 MPa was applied from both sides to have more homogeneous density compared to the classical dynamic compaction method. These samples were equilibrated in controlled relative humidity boxes starting from the highest relative humidity (*RH* = 97.3%, *s* = 3.8 MPa) towards the lowest relative humidity (*RH* = 9%, *s* = 331.3 MPa) to obtain the desorption path. In order to have data points for suction less than 3.8 MPa, the pressure plate method was used. Pressure plate tests make use of the axis translation technique. In these tests, a chamber is divided by a high air-entry pressure plate. Soil samples are placed on the top of the high air entry pressure plate, and the values of air pressure and water pressure are controlled. At the selected values of soil matric suction ($u_{a}-u_{w}$), samples are removed and weighed for the determination of water content and degree of saturation. The soil water retention curve obtained from these two tests is shown in Figure 2.

## 3. Theoretical Aspects of the Model

The numerical simulations on the drying and compression behavior of rammed earth walls were performed using CODE_BRIGHT (Olivella et al., 1996 [20]), which is a finite element 3-D program for the thermo-hydro-mechanical (THM) coupled analysis in geological media. This code works in combination with the pre/post-processor GiD, which was developed by the International Center for Numerical Methods in Engineering (CIMNE) at UPC Barcelona.

For the basic formulations in CODE_BRIGHT, a porous medium which is composed of grains, water, and gas is considered. The different aspects which are taken into account are thermal, hydraulic, and mechanical resolution, and the coupling between them. It uses a multi-phases and multi-species approach. The different phases that we consider here include solid phase (*s*): soil particles, liquid phase (*l*): water and dissolved air, and gas phase (*g*): dry air and water vapor. The different species include solid (*s*): same as solid phase, water (*w*): liquid water and evaporated in gas phase, and air: dry air and dissolved in liquid phase. More details on restrictions of the thermo-hydro-mechanical approach used in CODE_BRIGHT have been given in the PhD thesis of P. Chauhan [21]. However, some of the aspects and assumptions which are considered are mentioned below:The three phases are considered to be at the same temperature, i.e., a thermal equilibrium between phases.The various state variables (also called as unknowns) considered are solid displacements in 3 directions (*u*), liquid pressure ($P_{l}$), gas pressure ($P_{g}$), and temperature (*T*).Balance of momentum for the medium as a whole is reduced to the equation of stress equilibrium together with a mechanical constitutive model to relate stresses with strains. Strains are defined in terms of displacements.Small strains and small strain rates are assumed for solid deformation. Advective terms due to solid displacement are neglected after the formulation is transformed in terms of material derivatives.Balance of momentum for dissolved species and for fluid phases are reduced to constitutive equations (Fick’s law and Darcy’s law).Physical parameters in constitutive laws are function of pressure and temperature e.g., dynamic viscosity in Darcy’s law depends on temperature.

For writing the balance equations following notations were used: - $\phi_{p}$: porosity (-), ρ: density (kg/m${}^{3}$), j: total mass flux (*), i: non-advective mass flux (*), q: advective flux (*), u: solid displacements (m), σ: stress tensor (MPa), b: body forces (MN), ω: mass fraction (kg/kg), *e*: specific internal energy (J), $i_{c}$: conductive heat flux (*), $j_{e}$: energy flux for mass motion (*), $S_{l}$: liquid degree of saturation (-), $S_{g}$: gas degree of saturation (-).

Superscripts *w* and *a* refer to water and air, respectively. Subscripts *s*, *l* and *g* refer to solid, liquid and gas phase, respectively. (*) represents that the units of these flux terms depend on the type of flux such as point, line, surface, and volume flux.Assumptions and restrictions mentioned above have been used to write balance equations described in the following paragraph.

### 3.1. Balance Equations

The theoretical development of the balance equations is established by Olivella et al., 1996 [20] and has been summarized in this section. Regarding the balance equations; mass, momentum and energy conservation equations were used. The balance equations for mass consider an compositional approach which means that mass balance is performed for water, air, and solid species instead of phases. The total mass flux of a species in a phase (e.g., flux of water in gas phase $j_{g}^{w}$) is defined as the sum of three terms:Non-advective flux: $i_{g}^{w}$, i.e., diffusive/dispersive;Advective flux due to fluid motion: $\omega_{g}^{w}\rho_{g}q_{g}$, where $\omega_{g}^{w}$ is the mass fraction of water in gas phase, $\rho_{g}$ is gas density, and $q_{g}$ is the Darcy’s flux;Advective flux due to solid motion: $\omega_{g}^{w}\rho_{g}S_{g}\phi_{p}\frac{du}{dt}$, where $\frac{du}{dt}$ is the vector of solid velocities.


**Mass Balance of Solid**


The mass balance of solid present in the medium is presented as:(2)$$\frac{\partial (\rho_{s}\left(1-\phi_{p}\right))}{\partial t}+\nabla \cdot j_{s}=0$$
where, $\rho_{s}$ is mass of solid per unit volume of solid and $j_{s}$ is the flux of solid. The expression of porosity variation derived from Equation (Equation 2) is as following:(3)$$\frac{D_{s}\phi_{p}}{Dt}=\frac{\left(1-\phi_{p}\right)}{\rho_{s}}\frac{D_{s}}{D_{t}}+\left(1-\phi_{p}\right)\nabla \cdot \frac{du}{dt}$$
where, u is the solid displacement and $D_{s}/Dt$ is the material derivative in the Lagrangian framework. Thus, the porosity variation expressed in Equation (Equation 3) is due to the volumetric deformation and solid density variation.


**Mass Balance of Water**


The water exists in both liquid and gaseous phase; thus, the total mass balance of water is written as:(4)$$\frac{\partial (\left(\omega_{g}^{w}\rho_{g}S_{g}+\omega_{l}^{w}\rho_{l}S_{l}\right)\phi_{p})}{\partial t}+\nabla \cdot \left(j_{g}^{w}+j_{l}^{w}\right)=f^{w}$$
where, $\omega_{g}^{w}\rho_{g}$ is the mass of water in the gas phase, $\omega_{l}^{w}\rho_{l}$ is the mass of water in the liquid phase, and $f^{w}$ is the external supply of water. Using the material derivative following equation is obtained:(5)$$\begin{array}{c} \phi_{p}\frac{D_{s}\left(\omega_{g}^{w}\rho_{g}S_{g}+\omega_{l}^{w}\rho_{l}S_{l}\right)}{Dt}+\left(\omega_{g}^{w}\rho_{g}S_{g}+\omega_{l}^{w}\rho_{l}S_{l}\right)\frac{D_{s}\phi_{p}}{Dt}\\ +\left(\omega_{g}^{w}\rho_{g}S_{g}+\omega_{l}^{w}\rho_{l}S_{l}\right)\phi_{p}\nabla \cdot \frac{du}{dt}+\nabla \cdot \left(j_{l}^{{}^{\prime }w}+j_{g}^{{}^{\prime }w}\right)=f^{w} \end{array}$$
where, $j^{\prime }$ represents the sum of the non-advective and fluid motion advective terms i.e., excluding the solid motion advective terms from the total flux.


**Mass Balance of Air**


The mass balance equation for air is expressed in the same manner as the mass balance of water, taking into account the dry air as the main component of the gas phase and air dissolved in the liquid phase. The mass balance of air is written as:(6)$$\begin{array}{c} \phi_{p}\frac{D_{s}\left(\omega_{g}^{a}\rho_{g}S_{g}+\omega_{l}^{a}\rho_{l}S_{l}\right)}{Dt}+\left(\omega_{g}^{a}\rho_{g}S_{g}+\omega_{l}^{a}\rho_{l}S_{l}\right)\frac{D_{s}\phi_{p}}{Dt}\\ +\left(\omega_{g}^{a}\rho_{g}S_{g}+\omega_{l}^{a}\rho_{l}S_{l}\right)\phi_{p}\nabla \cdot \frac{du}{dt}+\nabla \cdot \left(j_{l}^{{}^{\prime }a}+j_{g}^{{}^{\prime }a}\right)=f^{a} \end{array}$$


**Momentum Balance of the Medium**


Momentum balance of the porous medium is reduced to stress equilibrium if the internal terms of stresses are neglected:(7)∇·σ+b=0
where, σ is the stress tensor and b represents the vector for body forces.


**Internal Energy Balance of the Porous Medium**


By taking into account the internal energy of each phase ($e_{s},e_{l},$ and $e_{g}$), the internal energy balance equation for the porous medium can be expressed as:(8)$$\begin{array}{c} \frac{\partial (e_{g}\rho_{s}\left(1-\phi_{p}\right)+e_{g}\rho_{g}S_{g}\phi_{p}+e_{l}\rho_{l}S_{l}\phi_{p})}{\partial t}-\frac{\phi_{p}S_{g}p_{g}}{\rho_{g}}\frac{\partial \rho_{g}}{\partial t}\\ +\nabla \cdot \left(i_{c}+j_{\mathrm{es}}+j_{\mathrm{eg}}+j_{\mathrm{el}}\right)=f^{Q} \end{array}$$
where, $i_{c}$ is the energy flux due to conduction through the porous medium, $j_{\mathrm{es}},j_{\mathrm{eg}}$, and $j_{\mathrm{el}}$ are the advective fluxes of energy by mass motion of every species in the medium, and $f^{Q}$ is an internal/external energy supply.

The various equilibrium equations and independent variables (unknowns) are summarized in Table 2.

### 3.2. Constitutive Equations

The various constitutive equations used are mentioned in detail in this section.


**Darcy’s Law**


The equation used for the liquid advective flux ($q_{l}$) was the Darcy’s law which is defined as:(9)$$q_{l}=-\frac{k_{i}k_{r}}{\mu_{l}}\left(\nabla P_{l}-\rho_{l}g\right)$$
where, $\mu_{l}$ (MPa.s) is the liquid dynamic viscosity, $P_{l}$ is the liquid pressure (MPa), $k_{i}$ is the intrinsic permeability, and $k_{r}$ is the relative liquid hydraulic conductivity.


**Retention Curve**


The retention behavior of rammed earth was defined by using the Van Genuchten 1980 [22] (VG) model. The Van Genuchten expression for effective degree of liquid saturation ($S_{el}$) is defined as:(10)$$S_{el}=\frac{S_{l}-S_{rl}}{S_{ml}-S_{rl}}={\left[\frac{1}{1+{\left(\alpha_{w}s\right)}^{n_{w}}}\right]}^{m_{w}}$$
where, $S_{l}$ is the actual degree of liquid saturation, $\alpha_{w}$ (MPa)${}^{-1}$, $n_{w}$, and $m_{w}$ are model parameters ($m_{w}$ is related to $n_{w}$ as $m_{w}=\left(n_{w}-1\right)/n_{w}$), and s is matric suction (MPa). Here, the maximum ($S_{ml}$) and residual ($S_{rl}$) degree of saturation were considered as 1 and 0 respectively, so $S_{el}=S_{l}$.

The unsaturated hydraulic conductivity function was defined as $K=k_{r}K_{sat}$, where $k_{r}$ is the relative hydraulic conductivity and $K_{sat}$ is the saturated hydraulic conductivity. The Mualem hydraulic conductivity model (1976) [23] using the Van Genuchten retention curve parameters was used to express $k_{r}$ as a function of $S_{l}$ coupled with Van Genuchten retention model, which gives:(11)$$k_{r}=\sqrt{S_{l}}{\left[1-{\left(1-S_{l}^{1/m_{w}}\right)}^{m_{w}}\right]}^{2}$$

The intrinsic permeability ($k_{i}$) can be related to the saturated hydraulic conductivity ($K_{sat}$) by the expression:(12)$$k_{i}=\frac{K_{sat}\cdot \mu_{l}}{\rho_{l}g}$$
where, $\rho_{l}$ is the density of water and $\mu_{l}$ is the dynamic viscosity of water.


**Fick’s Law**


The diffusive flux of water vapor in gas phase ($i_{g}^{w}$) was evaluated using Fick’s law of vapor diffusion:(13)$$i_{g}^{w}=-\left(t_{s}\phi_{p}\rho_{g}S_{g}D_{g}^{w}I\right)\nabla \omega_{g}^{w}$$
where, $t_{s}$ = tortuosity, $\phi_{p}$ = porosity, $\rho_{g}$ = gas density, $S_{g}=1-S_{l}$ is the gas degree of saturation, $D_{g}^{w}\left(m^{2}/s\right)$ is the diffusion coefficient of water vapor in the gas phase, and $\omega_{g}^{w}\left(\mathrm{kg}/\mathrm{kg}\right)$ is the mass fraction of water vapor in gas phase.

The molecular diffusion of vapor in gas phase is determined from the following equation:(14)$$D_{g}^{w}=D\left[\frac{{\left(273.15+T\right)}^{d}}{P_{g}}\right]$$
where, $P_{g}$ is the gas pressure in Pa, D (m${}^{2}$ s${}^{-1}$ K${}^{-2.3}$ Pa), and d are the vapor diffusion parameters.


**Fourier’s Law**


In order to determine the conductive flux of heat, Fourier’s law was used. Thermal conductivity was used in the Fourier’s law to compute the conductive heat flux $i_{c}$:(15)$$i_{c}=\lambda \nabla T$$
where, $\lambda \;(W\;m^{-1}\;K^{-1})$ is the thermal conductivity of the porous medium, which is further defined as:(16)$$\lambda =\lambda_{sat}\sqrt{S_{l}}+\lambda_{dry}\left(1-\sqrt{S_{l}}\right)$$
where, $\lambda_{dry}$ and $\lambda_{sat}$ are the thermal conductivity of the medium in the dry phase and saturated phase respectively. These parameters are further evaluated from the thermal conductivity of each phase taking into account the porosity of the medium as follows:(17)$$\lambda_{dry}=\lambda_{s}^{1-\phi_{p}}\lambda_{g}^{\phi_{p}}$$
(18)$$\lambda_{sat}=\lambda_{s}^{1-\phi_{p}}\lambda_{l}^{\phi_{p}}$$
where, $\phi_{p}$ is the porosity, and $\lambda_{s}$, $\lambda_{g}$, and $\lambda_{l}$$\left(W\;m^{-1}\;K^{-1}\right)$ are the thermal conductivity of the solid, gas and liquid phases respectively.


**Mechanical Constitutive Model**


The most suitable model that was found in CODE_BRIGHT was viscoplasticity model for unsaturated soils and rocks. This model, when combined with the linear elasticity model and adjusting the parameters used for viscosity in order to remove the viscous effects, a linear elastic–perfectly plastic model was obtained.

A linear elastic model is based on Hook’s law, which relates the total or effective stress to strains. It can be expressed using two soil parameters, modulus of elasticity (E) and Poisson’s ratio (ν). This model can be written in an incremental form as follows:(19)$$d\sigma =D^{e}\cdot d\varepsilon^{e}$$
where, dσ is the incremental stress tensor, $d\varepsilon^{e}$ is the incremental elastic strain tensor, and $D^{e}$ is the elastic stiffness matrix.

The elastic stresses are limited by plasticity. The yield function (F) and the associated plastic potential (G) are defined using the Drucker–Prager model (Figure 3) which is based on Mohr–Coulomb parameters:(20)G=F=q−Mp−k
with
(21)$$M=\frac{6sin\phi^{\prime }}{3-sin\phi^{\prime }}\;\mathrm{and}\;k=Mc/tan\phi^{\prime }$$
where, *c* is the apparent cohesion, which is defined in terms of effective cohesion *c* and suction *s*, and $\phi^{\prime }$ is the effective friction angle. *p* (mean stress) and *q* (deviatoric stress) are related to the first invariant of stress tensor and second invariant of deviatoric stress tensor respectively which are defined as:(22)$$p=\frac{1}{3}I_{1}=\frac{1}{3}\left(\sigma_{xx}+\sigma_{yy}+\sigma_{zz}\right)$$
(23)$$q=\frac{1}{\sqrt{2}}\sqrt{{\left(\sigma_{xx}-\sigma_{yy}\right)}^{2}+{\left(\sigma_{yy}-\sigma_{zz}\right)}^{2}+{\left(\sigma_{zz}-\sigma_{xx}\right)}^{2}+6\left(\tau_{xy}^{2}+\tau_{yz}^{2}+\tau_{zx}^{2}\right)}$$

For the constitutive modeling of unsaturated soil, it is essential to incorporate the effect of suction in the failure criterion. In unsaturated soil mechanics, the stress state of a porous medium can be represented by two approaches. The first considers two independent stress variables such as net stress ($\sigma -u_{a}I$) and matric suction ($u_{a}-u_{w}$), which are measurable and have an experimental significance. On the other hand, an approach that uses single effective stress to define the stress state of a multi-phase porous medium can also be used. Effective stress is the stress that is being transferred by grain to grain contact and responsible for the mobilization of shear strength in the soil. In order to represent the results at different suction conditions in a single stress framework, Bishop’s generalized effective stress [24] was used in this study:(24)$$\sigma^{\prime }=\sigma -u_{a}I+\chi sI$$
where, $\sigma^{\prime }$ is the effective stress tensor, σ is the net stress tensor, s is the suction, I is the Identity matrix and χ is the effective stress parameter which is a function of degree of saturation. The shear strength expression in terms of normal effective stress ($\sigma_{n}^{\prime }$) can be written as:(25)$$\tau =c^{\prime }+\sigma_{n}^{\prime }tan\phi^{\prime }$$

Using the Bishop’s effective stress approach in the shear strength expression, we obtain:(26)$$\tau =c^{\prime }+s\chi tan\phi^{\prime }+\left(\sigma_{n}-u_{a}\right)tan\phi^{\prime }$$

Thus the apparent cohesion *c* can be defined as:(27)$$c=c^{\prime }+s\chi tan\phi^{\prime }$$

It is to be noted that the friction angle is not affected by the variation of suction in the model.

### 3.3. THM Coupling

Let us discuss the THM coupling between the mechanical behavior, liquid water and vapor transfer, and heat transfer shown in Figure 4.

Regarding the hydro-mechanical coupling, the mechanical effect on the liquid water transfer is taken into account using Darcy’s law, as a change in porosity will lead to change in unsaturated permeability. The hydric effect on the mechanical behavior is considered using effective stress (depending on suction). The liquid water and vapor transfer are linked one to the other with the soil water retention curve and Kelvin’s equation.

For the hydro-thermal coupling, the thermal effect on the vapor transfer is considered using Fick’s law where the diffusion coefficient is dependent on temperature. In contrast, the hydric effect on the thermal behavior is considered using Fourier’s law, where the thermal conductivity is dependent on the liquid degree of saturation.

Finally, for the thermo-mechanical coupling, the mechanical effect on the heat transfer in considered using Fourier’s law, as thermal conductivity is dependent on porosity. In addition, there is no direct coupling on the thermal effect on mechanical behavior since the mechanical model is temperature independent. Thus, suction is the link between hydro-mechanical and hydro-thermal coupling.

## 4. Material Parameters

### 4.1. Hygric, Hydric and Thermal Parameters

As suction is the link between hydro-mechanical and hydro-thermal coupling, the most important parameters include the retention curve parameters. The Van-Genuchten (VG) model [22] for the soil–water retention curve previously described in Equation (Equation 10) was used for the retention behavior (Figure 5). The VG model parameters were evaluated as $\alpha_{w}$ = 3.479, $n_{w}$ = 1.379, and $m_{w}$ = 0.2748. All these parameters have been obtained from the experimental measurements on representative samples. Soil-water retention curve has been obtained from pressure plate and saline solution method. For the saline solution method, 5 samples were equilibrated at 7 different suction states starting from lower values of suction (3.8 MPa) to higher values of suction (331.3 MPa) mentioned in Table 1. Experimental results are plotted on the curve of Figure 5. The details of the experimental procedure are described in the PhD thesis of P. Chauhan [21]. At the compaction conditions, the degree of saturation determined experimentally was 0.805 (corresponding to *w* = 12.5 %). Corresponding to this hydric state, the initial suction at compaction was determined from the fitted retention curve as 0.328 MPa.

The unsaturated hydraulic conductivity (*K*) which was defined previously as the product of saturated hydraulic conductivity ($K_{sat}$) and relative hydraulic conductivity ($k_{r}$) was modeled using Mualem hydraulic conductivity model which uses the parameter $m_{w}$ of the Van-Genuchten retention curve model. The variation of relative hydraulic conductivity ($k_{r}$) evaluated using Equation (Equation 11) as a function of liquid degree of saturation is shown in Figure 6.

The saturated hydraulic conductivity was evaluated from the rammed earth walls by drilling it in the direction of drying, i.e., parallel to the layers. These samples were tested in a triaxial device. The average saturated hydraulic conductivity was determined equal to 6.3 × $10^{-9}$ m/s, and the intrinsic permeability was deduced using Equation (Equation 12) as 5.7 × $10^{-16}$ m${}^{2}$.

The range of thermal conductivity (λ) of rammed earth in its dry state recommended according to“Guide de Bonnes Pratiques (GBP) Pisé, 2018“ [15] is from 0.46 to 0.81 W m${}^{-1}$ K${}^{-1}$. In the model used here, the values of thermal conductivity for each phase, i.e., solid, liquid, and gas phase are used, and the thermal conductivity of the porous medium is evaluated based on the degree of saturation. The thermal conductivity of the liquid ($\lambda_{liq}$) and gas ($\lambda_{gas}$) phase taken as thermal conductivity of water and air is 0.6 and 0.025 W m${}^{-1}$ K${}^{-1}$ respectively. The thermal conductivity of solid phase ($\lambda_{solid}$) based on the range specified before was taken as 1.5 W m${}^{-1}$ K${}^{-1}$.

The values of the vapor diffusion parameters are taken as the following: *D* = 5.9 × $10^{-6}$ m${}^{2}$ s${}^{-1}$ K${}^{-2.3}$ Pa, *d* = 2.3, and tortuosity $t_{s}$ = 1. These are the classical values for water vapor diffusion taken from the CODE_BRIGHT manual [25]. All the hydro-thermal parameters used are summarized in Table 3.

### 4.2. Mechanical Parameters

For the elastic part, the values of modulus of elasticity and Poisson’s ratio (ν) were taken as constant equal to 150 GPa and 0.25 respectively, from the literature [9,12]. For the plasticity part, the Drucker–Prager failure surface, which is based on Mohr–Coulomb parameters, was adopted (Equation (Equation 20)). The values of slope and intercept of the failure envelope in terms of effective stresses ($p^{\prime }$–*q*) were determined to be *k* = 0.0921 MPa and *M* = 1.3105. These values were determined from the 4 consolidated undrained (CU) triaxial tests carried out on saturated small cylindrical samples in experimental study [16]. The values of effective cohesion $c^{\prime }$ and effective friction angle $\phi^{\prime }$ were determined to be 43.9 kPa and 32.5${}^{\circ }$.

The expression of Bishop’s stress was taken into account in the expression of shear strength (Equation (Equation 26)). To define this, the effective stress parameter χ, which is a function of the liquid degree of saturation ($\chi =S_{l}^{\alpha }$) needs to be determined. 21 The unconfined compressive strength and 21 unsaturated triaxial test were performed at 7 different initial suction states in the experimental study on the same material. The details of the experimental procedure are described in Ph.D. thesis of P. Chauhan [21] and Chauhan et al., 2019 [16]. The value of the exponent was determined to be α=1.9081.

Since the parameter α affects the effective stress behavior, a sensitivity analysis was performed. The value of α for fitting with the experimental data was 1.9081 with an R-squared value of 0.88. Thus, an interval of 10% was chosen and the sensitivity analysis for the effective stress parameter χ was done for α±5%α i.e., between α=1.8127 and α=2.0035. Figure 7 shows the variation of compressive strength obtained for the different values of effective stress parameter and compared with the experimental tests on representative samples [21]. It can be seen that, for a small change in the value of χ, which is a function of liquid degree of saturation ($\chi =S_{l}^{\alpha }$), there is a very significant change in the value of compressive strength.

Thus, the compressive phase simulations are very sensitive to the effective stress parameter χ. The compressive strength from the simulations for α=2.0035 seems to be very close to the experimental results except at the initial phase of drying. Finally, the best compromise for α was 1.9081 i.e., the middle-dashed line on Figure 7 by lack of better experimental results at higher range of degree of saturation. Thus, more tests should be done near the compaction degree of saturation to determine α and hence χ accurately. For this reason the sensitivity analysis was performed only on α parameter and detailed in the Ph.D. thesis of P. Chauhan [21].

The value of porosity ($\phi_{p}$) was taken equal to 0.291 from a previous study on rammed earth columns using dynamic compaction (Chauhan et al., 2020 [26]). The value of all the mechanical parameters is summarized in Table 3.

## 5. General Considerations for the Simulations

### 5.1. Definition of a Safety Failure Envelope

The Drucker–Prager failure envelope described in the previous section is shown in Figure 8 and is labeled as an ultimate failure surface. The equation for this failure surface is as follows:(28)$$q=k+Mp^{\prime }$$
where, *k* = 0.09021 MPa, and *M* = 1.3105 as already described in the previous section (Section 4.2), *q* is the deviatoric stress, and $p^{\prime }$ is the mean effective stress defined as:(29)$$p^{\prime }=p-u_{a}+\chi s$$

In the practical guide used for construction using rammed earth in France called “Guide de Bonnes Pratiques (GBP) Pisé, 2018” [15], it is mentioned that “the stresses in the wall should not exceed 0.2 MPa or the one-third of the failure stress of the rammed earth”. In a similar approach, a safety factor is proposed in this contribution in order not to reach the ultimate failure surface. This safety failure envelope is chosen such that the angle of the failure envelope ($\alpha_{f}$) is taken as one-third of the ultimate failure surface, i.e., $\alpha_{f}=\beta_{f}/3$ (Figure 8). This failure envelope is the central third cone of the original Drucker-Prager failure surface in the 3D principal stress plane. This arbitrary choice allows to reduce in the same way the cohesion and the friction angle. The equation for this failure surface is as follows:(30)$$q=k_{s}+M_{s}p^{\prime }$$
where, $k_{s}=0.0217$ MPa and $M_{s}=0.3163$, which were determined from the geometrical relation between the failure envelopes. It has to be noticed that the relevance of this safety failure envelope choice could be enhanced by further studies.

These hypotheses could lead to an estimation of the drying time for safety. The safety failure envelope will be further used to study the mechanical behavior of rammed earth walls and to determine the appropriate time of drying required for subsequent floor construction.

### 5.2. Environmental Conditions

Two different environmental conditions are considered for performing the drying simulations, i.e., a ”summer-like“ and ”winter-like“ conditions. These conditions are considered based on the temperature and relative humidity values at Le Bouget-du-Lac region in France (Figure 9) for the year 2019. There is no significant variation of the average atmospheric conditions in the past 3 years (2018–2020), and the year 2019 was chosen arbitrarily.

In summer-like conditions, the temperature and RH are taken as 20 °C and 60% respectively. These are the average values during the summer months (June–August), with a weak variability. In winter-like conditions, the temperature and RH are taken as 5 °C and 85% respectively. These are the average values during the winter months (November–January). These values are only an estimation since it is not possible to take a single value for different months. In addition, our simulations does not take into account freezing during winter because it does not make sense with respect to drying. The suction values corresponding to these conditions are also mentioned in Table 4.

Regarding the hydraulic boundary conditions, when an imposed liquid pressure (and hence suction) is applied as the boundary condition, the boundary immediately reaches the final suction state and is supposed to be in equilibrium with the atmosphere. This condition is not realistic since the equilibrium at the surface is reached over time, and the boundary reaches the final suction state asymptotically. Thus, in this study, a more realistic atmospheric boundary condition was used. CODE_BRIGHT allows imposing a boundary condition that includes mass and heat conditions to be applied in terms of atmospheric data.

Atmospheric boundary condition allows the application of boundary conditions in terms of evaporation, rainfall, radiation, and heat exchanges. In this way, it is able to simulate the complex soil–atmosphere interactions. The flux boundary condition was used to express these phenomena for the three different components, i.e., water, air, and energy in terms of their respective state variables, i.e., liquid pressure, gas pressure, and temperature or dependent variables (such as liquid saturation degree, fraction of water in gas phase as water vapor). In addition, it uses various ambient conditions such as relative humidity, atmospheric gas pressure, ambient temperature, and air velocity.

In CODE_BRIGHT, the evaporation flux $E_{v}$ ($\mathrm{kg}\;s^{-1}\;m^{-2}$) is defined by an aerodynamic diffusion relation:(31)$$E_{v}=\frac{V_{k}v_{a}s_{f}}{{\left(ln\frac{z_{a}}{z_{o}}\right)}^{2}}\left(\rho_{va}-\rho_{v}\right)$$

The evaporation flux can also be defined as [27]
(32)$$E_{v}=h_{m}\left(\rho_{va}-\rho_{v}\right)$$
where, $\rho_{va}$ and $\rho_{v}$ are the absolute humidity of the atmosphere and at the boundary respectively, $V_{k}$ is the von Karman’s constant, $s_{f}$ is a stability factor, $v_{a}$ is the wind velocity, $z_{o}$ is the roughness length, $z_{a}$ is the screen height at which $v_{a}$ and $\rho_{va}$ are measured and $h_{m}$ (m/s) is the surface mass transfer coefficient. Thus, by equating Equations (Equation 31) and (Equation 32) we have:(33)$$h_{m}=\frac{kv_{a}s_{f}}{{\left(ln\frac{z_{a}}{z_{o}}\right)}^{2}}$$

Thus we can use arbitrary values of the parameters on the right side of Equation (Equation 33) to implement the value of surface mass transfer coefficient ($h_{m}$), and thus, it is the key parameter instead of air velocity ($v_{a}$). The surface mass transfer coefficient ($h_{m}$) can be evaluated from the heat transfer coefficient $h_{c}$ (W m${}^{-2}$ K${}^{-1}$) using the Lewis relation:(34)$$h_{m}=\frac{h_{c}}{\rho_{a}C_{p}}$$
where, $\rho_{a}$ is the air density = 1.223 kg/m${}^{3}$ and $C_{p}$ is the air specific heat = 1.006 kJ kg${}^{-1}$ K${}^{-1}$ at T = 15.5 °C.

In the simulations, the value of surface heat transfer coefficient ($h_{c}$) is taken as 25 W m${}^{-2}$ K${}^{-1}$, which is corresponding to a value of surface mass transfer coefficient ($h_{m}$) of 0.02 m/s. This value is suggested by Réglementation Thermique 2012 [28] for drying in an external environment in France. This recommended value is slightly low with respect to wind condition. All the different values used for the atmospheric boundary conditions used are mentioned in Table 4.

## 6. Case Study: THM Simulations on Single Wall

THM coupled numerical simulations of rammed earth wall were performed using CODE _BRIGHT. The gas phase in the model was considered immobile, and atmospheric gas pressure was considered ($P_{g}=$ 0.1 MPa). The 3D geometrical model of the wall considered has a length and height of 3 m, and a width of 0.45 m which is a classical width for rammed earth walls in France (Figure 10). Since no displacements were allowed in the out-of-plane direction (*X* direction), a 2D model is sufficient with a plane strain condition. A 2D mesh of 12 × 10 (120) quadrilateral linear (Q4) elements was used.

Regarding the mechanical boundary conditions considered for these simulations, the bottom surface was fixed ($u_{y}=u_{z}=0$, the axes being represented in Figure 10). On the two lateral surfaces (perpendicular to the *x*-direction), displacement in the perpendicular directions was not allowed ($u_{x}=0$). The two surfaces perpendicular to the *z*-direction were stress-free with displacements being allowed. Vertical stress will be applied on the top surface.

### 6.1. Compression of the Wall at Compaction Hydric State

In this part, a single mechanical simulation without drying was carried out by applying vertical stress gradually at the top surface in order to determine the load sustained by the wall before any part of the wall reaches plasticity. Both the ultimate and safety failure surface was considered. The initial hydric state of the wall was corresponding to the ‘as compacted state’ of the material, i.e., *s* = 0.328 MPa and water content of 12.5%. Since it is a mechanical only simulation, no hydric boundary conditions were required.

In order to determine the critical point for the plastic failure of the wall, stress paths were studied at different distances from the bottom of the wall.The stress paths of various points in the wall were analyzed and the point for which the stress path was steepest and reaches failure first was labeled as the “critical point”. Figure 11 shows the stress path of the most critical point during the gradual loading of the wall, which is at 0.5 m from the bottom and at the middle of the wall (since it will be the wettest once drying will be initiated). The stress path for this point begins at the initial suction state where even though the wall is free of stress at its boundary, the internal stress is not zero but depends on the water retention conditions through the product between the suction and effective stress parameter χ (Equation (Equation 29)). In other words, suction provides internal confining stress to the wall.

The stress state due to the self-weight of the wall at the critical point, is shown in Figure 11. Once the stress is applied on the top surface gradually, the stress path evolves linearly, and the critical point reaches the safety failure surface at 0.07 MPa vertical stress and ultimate failure envelope at 0.94 MPa vertical stress.

According to GBP, Pisé [15], the vertical stress to be considered for subsequent floor construction can be taken as 0.2 MPa, taking into account the self-weight of the walls in the subsequent floor, floor loading, and other live loads. The detailed calculation of this value of stress is shown in Appendix A. It can be seen from the stress path that at 0.2 MPa stress, the wall is very safe with regards to the ultimate failure surface. On the other hand, the stress state is outside the safety failure surface. Thus it can be inferred that the safety criterion suggested by the practical guide is very conservative. It is to be noted that in this simulation, the aim is not to reach a specific amount of vertical stress, but to analyze the failure stress for the plastic failure criterion, the safety failure criterion, and the GBP criterion (0.2 MPa of vertical stress).

In the further simulations, the walls will be subjected to drying for different durations, and after the drying period, vertical stress of 0.2 MPa will be applied from the top. The drying period will be determined when the critical point is safe with regards to the safety failure envelope.

### 6.2. Drying in Summer-like Conditions

These simulations will be carried out in two phases. In the first phase, drying will be imposed on the two lateral surfaces as shown in Figure 10 using the atmospheric boundary conditions for the summer-like environment (Table 4) at different duration. In the second phase, the wall is compressed by gradually increasing the vertical stress from the self-weight stress state up to an increment of 0.2 MPa at the top boundary. The safety of the same critical point mentioned previously with regards to the safety failure envelope will be analyzed.

Figure 12 shows the stress path of the critical point at 0, 30, 60, 90, and 120 days of drying. Firstly, from the initial compaction suction state, the self-weight is applied on the wall. Due to drying the stress path translates to the right by an amount χs in the $p^{\prime }$–*q* plane, which depends on the suction and degree of saturation. After the first 30 days of drying, the stress path is significantly shifted to the right. The suction value at the critical point increases while the degree of saturation decreases but the product of $S_{l}^{1.9081}s$ increases. Still, upon the stress application of 0.2 MPa, the stress path ends above the safety failure surface. This drying process is continued for another duration until the stress state lies below the safety failure surface. The effect of drying on the stress path is non-linear since the increase in χs decreases at the drying intervals of 30 days.

The isochrones of suction and degree of saturation across the width of the rammed earth wall have been shown in Figure 13. From the suction profiles, it can be seen that suction at the boundary increases gradually over time and approaches the final equilibrium value of 69.13 MPa (Table 4). At day 0 of drying, the wall is at a uniform suction state of 0.328 MPa ($S_{l}=0.805$), while at other days, the wall has non-uniform suction state. At 30 days of drying, the suction at the center of the wall reaches to suction of 1.4 MPa and degree of saturation of 53 %. Thus, a significant amount of drying has taken place within the first 30 days. Finally, at 120 days of drying, the suction at the center of the wall reaches 3.53 MPa with degree of saturation of 38 %. The stress path after 120 days of drying is the first to lay under the safety criterion.

Finally, the simulations shows that it takes between 3–4 months of drying for the critical point in the wall to be safe. In addition, this period of drying is also recommended in the practical guide [15]. This drying period at the same value of relative humidity and temperature is reasonably possible between the months of June and September, based on the actual weather data shown in Figure 9. Thus, it takes a significant amount of time for the single wall to dry sufficiently in summer-like conditions in order to construct another floor above it by taking into account the safety criterion suggested in the practical guide. It is to be noted that there can be different reasons for this estimated long drying period:In these simulations, the solar radiation is not taken into account, which can affect the drying duration.The surface mass/heat transfer coefficient value needs to be evaluated for the actual conditions taking into account the air velocity profile.The choice of failure surface recommended by the practical guide might be very conservative. This choice is arbitrary and could be more adapted in later simulations, according to a comparison between the field guidelines and the simulations.Finally, the failure criterion has been determined at a small scale samples, with double compacted material, and sieved earth, which may not be very representative of the real structure.

### 6.3. Drying in Winter-like Conditions

In these simulations, the drying was carried out in winter-like environmental conditions, as mentioned in Table 4. The stress path for the drying and stress application is shown in Figure 14. Due to the higher relative humidity and lower temperature, the drying process is slower compared to summer-like conditions and the increase in mean effective stress due to drying is significantly reduced. The effect of drying on the stress path can be seen in Figure 14, where the stress path is evolving non-linearly at 30 days drying intervals. The drying time required for the critical point to be safe is around 180 days.

The suction and degree of saturation isochrones along the width of the rammed earth wall are shown in Figure 15. The suction profiles show that the suction at the boundary increases gradually over time and approaches the final equilibrium value of 20.86 MPa (Table 4). This equilibrium at the surface takes more time than in summer-like condition (Figure 13). After 30 days of drying, the center of the wall reaches the suction of 1.08 MPa and degree of saturation of 58%. Finally, at 180 days of drying (6 months), the center of wall reaches suction of 3.32 MPa and degree of saturation of 39%. A comparison between the degree of saturation profiles with summer-like conditions can be made Figure 13). The isochrones in the summer-like conditions start to flatten close to the boundary at around 30 days, whereas in winter-like conditions, the whole curve is mostly concave downwards at 180 days of drying.

The time for the drying simulations based on the winter-like environmental conditions considered is longer than the actual winter months shown in Figure 9. Thus, it is not possible to dry the wall in these conditions. In the practical guide, it is mentioned that it is not reasonable to build in winter conditions not only because of freezing risk, but also because of the excessive time of drying. Hence another point of improvement would be to take the real-time weather data instead of a constant value of relative humidity and temperature. In addition, all the points discussed for the drying in summer-like conditions (Section 6.2) such as the solar radiation, surface heat/mass transfer coefficient, and the choice of failure criterion can be improved to get more accurate values of for the drying period.

The suction evolution at the center of the wall is also compared in Figure 16. The time lag between the summer-like and winter-like conditions is around 10 days for 1 MPa suction, 26 days for 2 MPa suction, and 56 days for 3 MPa suction. This time lag is gradually increasing for the duration of drying studied.

## 7. Conclusions and Perspectives

In this study, a methodology was developed in order to determine the drying period required for the walls to gain enough strength in order to build the subsequent floor, which is an important practical issue in rammed earth construction. In this regards, THM-coupled simulations of two different configurations of walls were carried out at summer-like and winter-like environmental conditions while taking into account a safety criterion using a similar approach from ‘Guide des Bonnes Pratiques de construction en Pisé’ [15]. It was decided to implement the unsaturated soil mechanics approach using suction as the state variable. Suction was used to describe the hydric state of the wall and was further linked with the thermal exchanges and the mechanical behavior. Thus it served as a link between the hydro-thermal and hydro-mechanical coupling phenomenon. The importance of using unsaturated soil mechanics approach in the domain of rammed earth construction is thus highlighted.The simulations using CODE_BRIGHT takes into account the evolution of the mean effective stress as a function of suction. Nevertheless, this approach remains limited in particular because it does not take into account the deformation evolution of the medium. This modeling is therefore on the conservative side.

For the drying of the walls, atmospheric boundary conditions were used during the modeling. In these conditions, the boundary reaches the final suction state gradually, which is more realistic than the imposed suction conditions.

The drying periods were determined by analyzing the stress paths of the critical point on application of 0.2 MPa vertical stress at top of the wall. The stress paths were dependent on the suction state of the critical point. Upon drying the suction state increased from 0.33 MPa to around 3.5 MPa. Using the effective stress formulation, the mean effective stress reached a value of 0.67 MPa corresponding to a deviatoric stress of 0.22 MPa and consequently making it safe with regards to safety failure surface. For a single wall subjected to drying in summer-like conditions, a drying period between 3–4 months is needed according to our results. According to the guide of good practices [15], a drying time of 3 to 4 months after manufacturing is recommended, which is in good agreement with our simulations. Whereas, in the winter-like conditions, a drying period of around 6 months was needed according to our results and our weather conditions. This seems not to be reasonable cause winter period in Le Bourget-du-Lac is not so long. In addition, the guide of good practices indicates that it is not reasonable to build in winter [15]. Thus, the recommendations are in good agreement with our simulations. The drying period determined from the simulations corresponds to the drying period suggested in the guidelines. The time lag of suction evolution between the summer-like and winter-like condition increased gradually from 0 days for 1MPa suction, 26 days for 2 MPa suction, and 56 days for 3 MPa suction. Since the stress state of the wall is very far from the ultimate failure criterion for the material, the safety criterion is too conservative. Therefore, it is possible to suggest a less conservative safety criterion without compromising the safety factor significantly.

Thus, our approach makes it possible to objectify the conditions for drying a rammed earth wall. This work can be further enhanced by the different perspectives which are mentioned as follows:In these simulations, solar radiation and rainfall were not considered, which can be taken into account to improve the analysis.The surface heat/mass transfer coefficient value was taken from the literature, which was slightly conservative in nature. Thus, it can be determined for the actual conditions by taking into account the wind velocity.Since it was observed that the drying period was higher than 4 months of the actual summer-like/winter-like conditions, which is not practical, it is recommended to use actual time-varying weather conditions.A less conservative safety criterion can be proposed without significantly compromising the safety factor. In this way, the drying duration required will be significantly reduced while still being far from the ultimate failure state.The failure criterion was determined on small samples, sieved at 5mm, double compacted, and are thus not very representative of the behavior of real structures, with more variability in density and granulometry.

## Figures and Tables

**Figure 1 materials-15-00362-f001:**
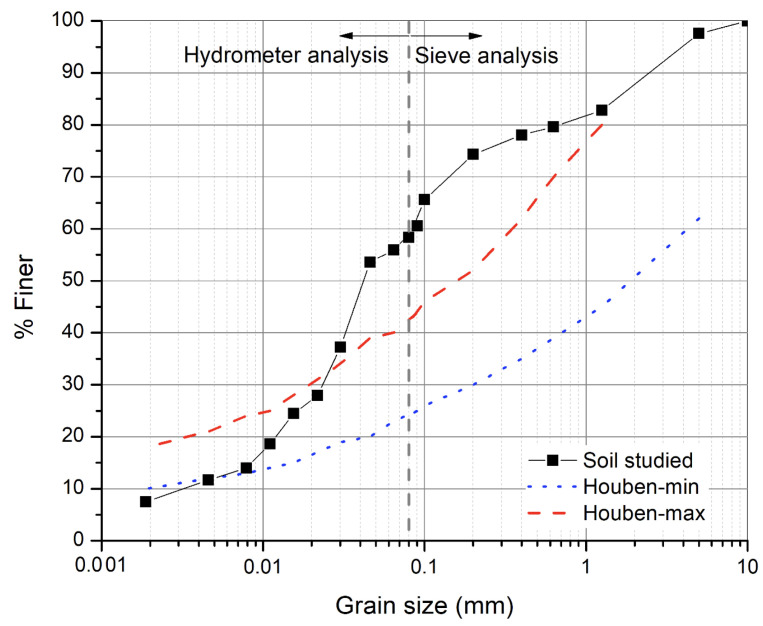
Particle size distribution of the soil along with the limit range specified by Houben et al., 1994 [18].

**Figure 2 materials-15-00362-f002:**
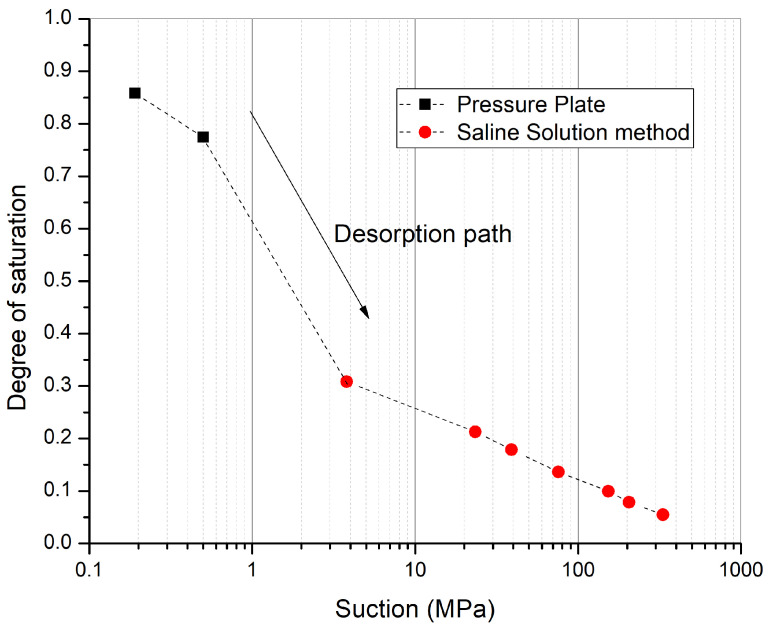
Soil water retention curve using the results of the pressure plate test and saline solution method.

**Figure 3 materials-15-00362-f003:**
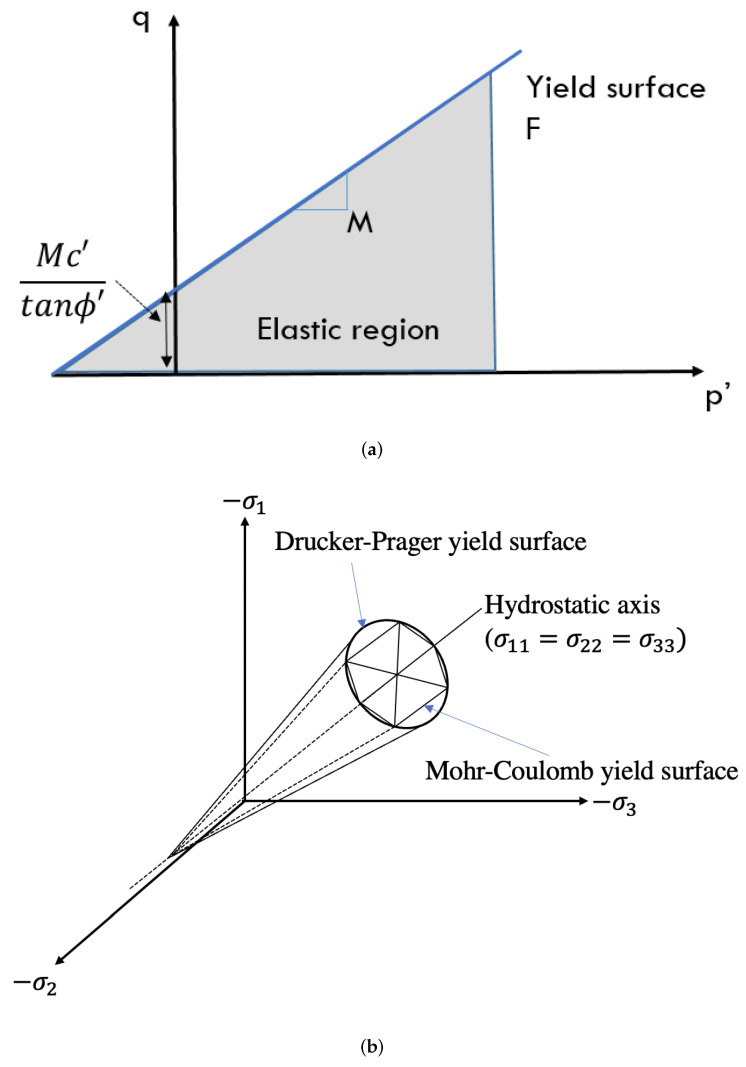
Yield surface for Drucker-Prager plasticity model in p’q space (**a**) and in principal stress space (**b**).

**Figure 4 materials-15-00362-f004:**
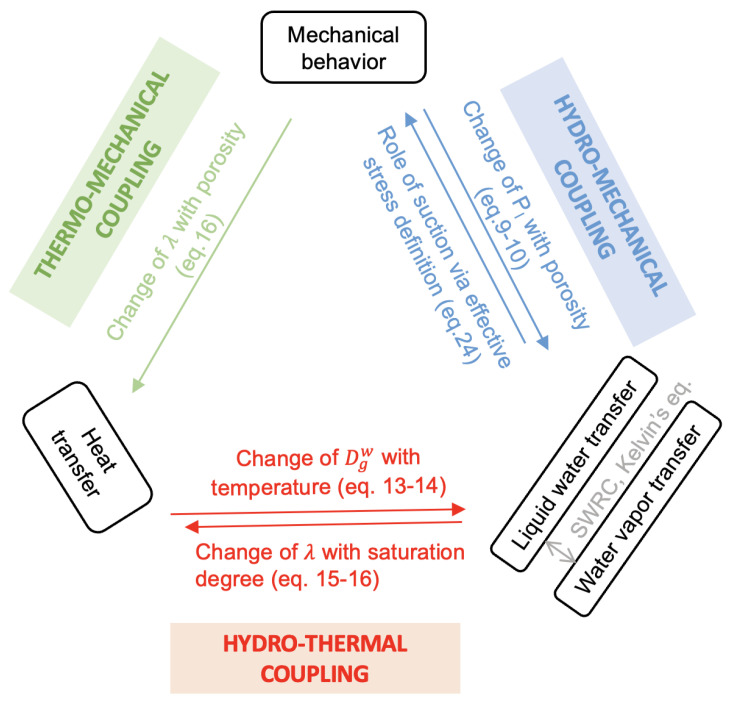
THM coupling between the mechanical behavior, liquid water and vapor transfer, and heat transfer.

**Figure 5 materials-15-00362-f005:**
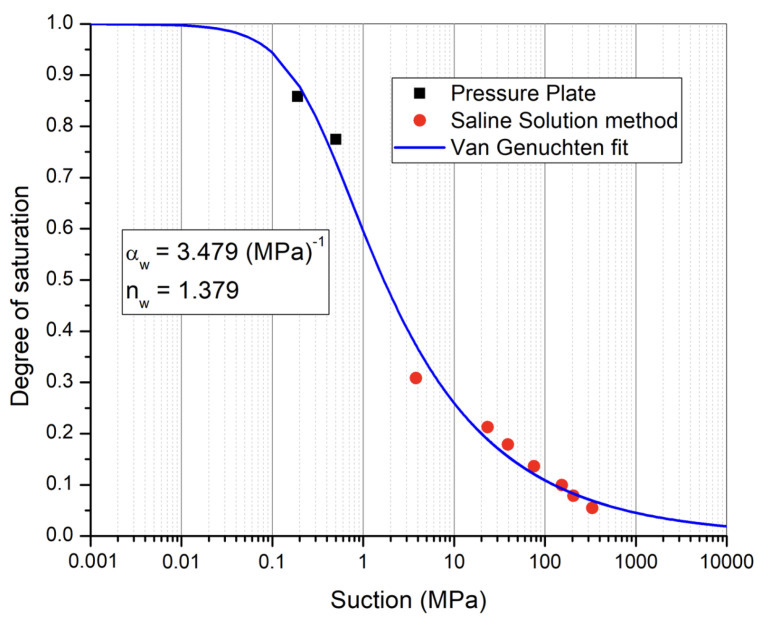
Soil water retention curve obtained from pressure plate and saline solution method using Van-Genuchten fitting parameters.

**Figure 6 materials-15-00362-f006:**
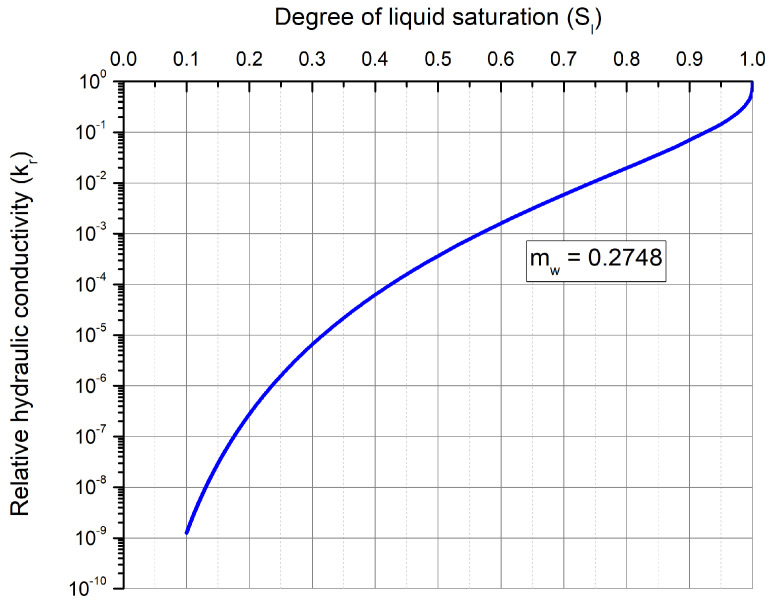
Relative hydraulic conductivity as a function of liquid degree of saturation using Mualem model [23].

**Figure 7 materials-15-00362-f007:**
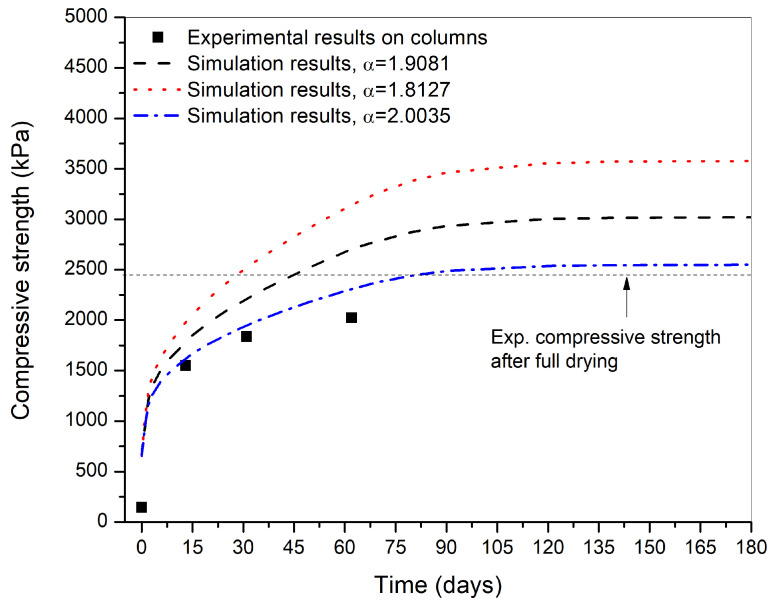
The variation of compressive strength with drying duration for the sensitivity analysis performed on the effective stress parameter χ.

**Figure 8 materials-15-00362-f008:**
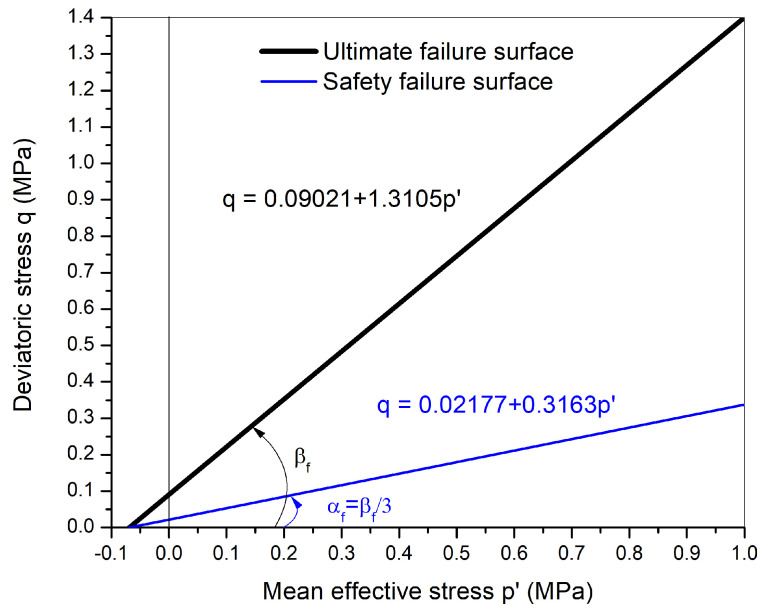
The ultimate failure envelope and safety failure envelope in $p^{\prime }$-*q* plane.

**Figure 9 materials-15-00362-f009:**
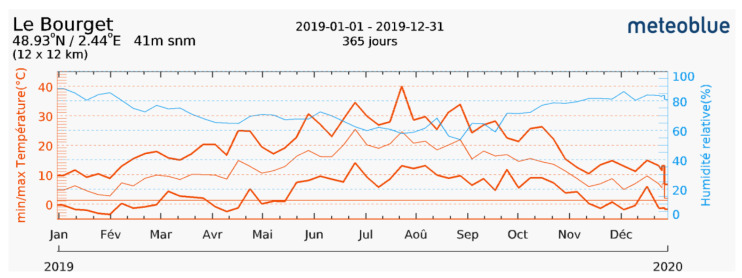
The weather data related to temperature and relative humidity for Le Bouget-du-Lac (year 2019) taken from www.meteoblue.com (accessed on 31 December 2020).

**Figure 10 materials-15-00362-f010:**
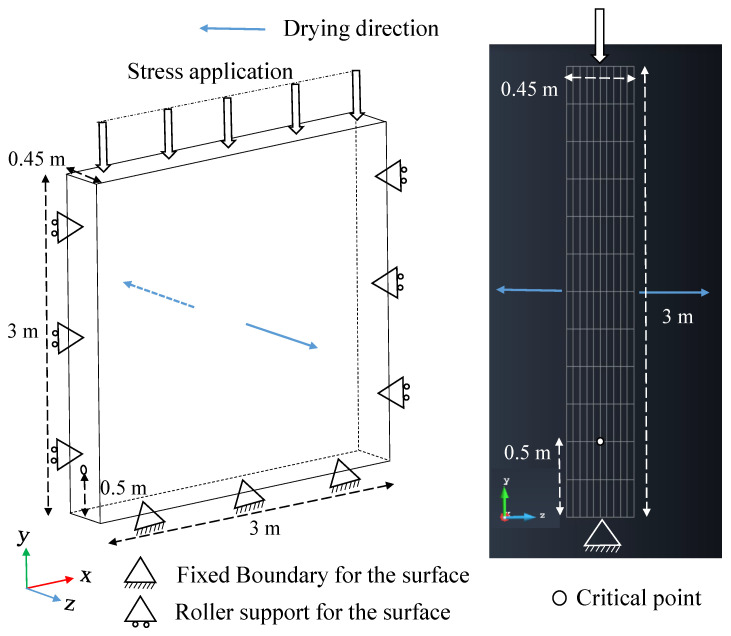
The geometrical model (3D) of the problem (**left**) and the 2D mesh composition (**right**) showing the drying direction and loading.

**Figure 11 materials-15-00362-f011:**
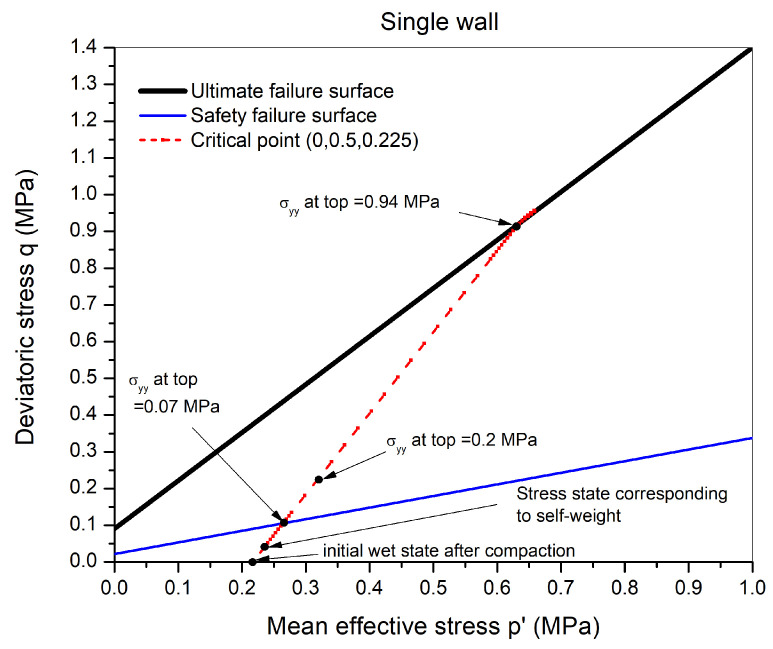
Stress path for single wall at the critical point due to vertical stress at top starting from optimum compaction hydric state.

**Figure 12 materials-15-00362-f012:**
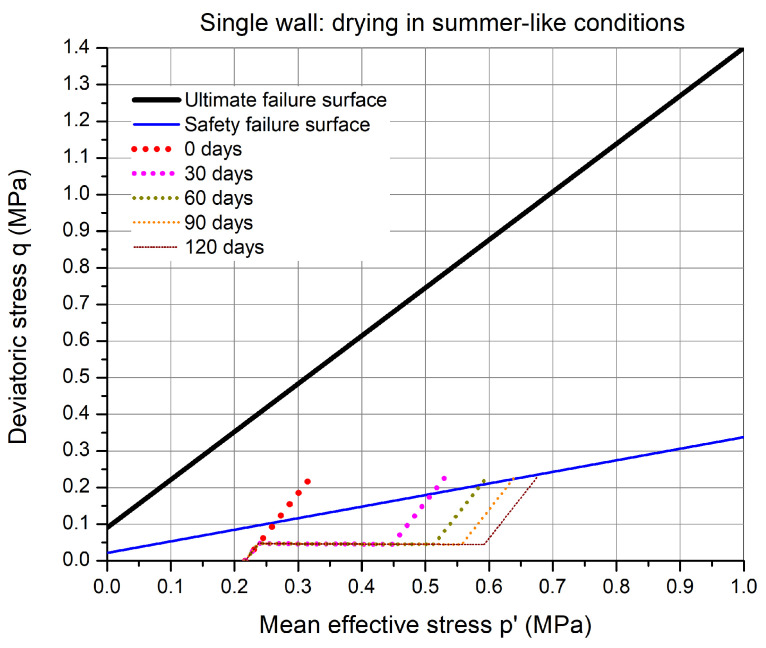
The stress path in the $p^{\prime }$–*q* plane for the critical point in the wall at different duration of drying in summer-like environmental conditions.

**Figure 13 materials-15-00362-f013:**
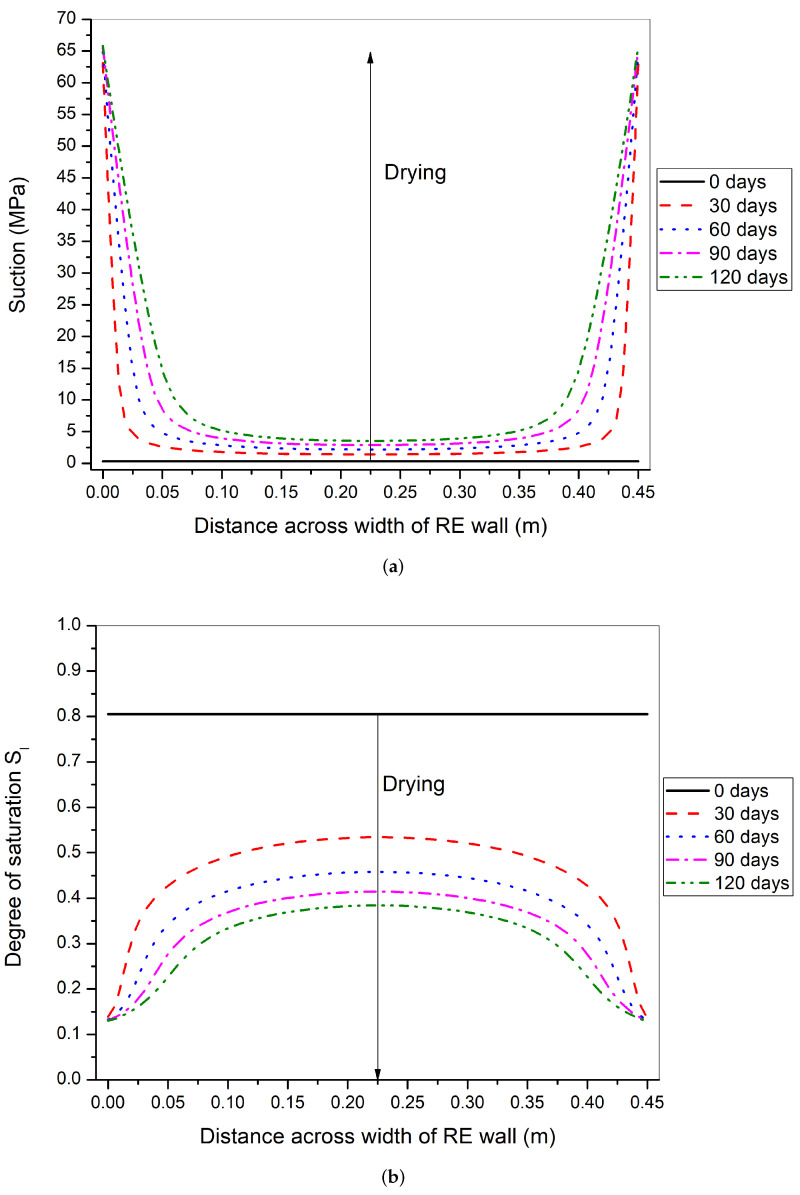
Suction (**a**) and degree of saturation (**b**) variation across the width of rammed earth wall at 0, 30, 60, 90, and 120 days of drying period in summer-like environmental conditions.

**Figure 14 materials-15-00362-f014:**
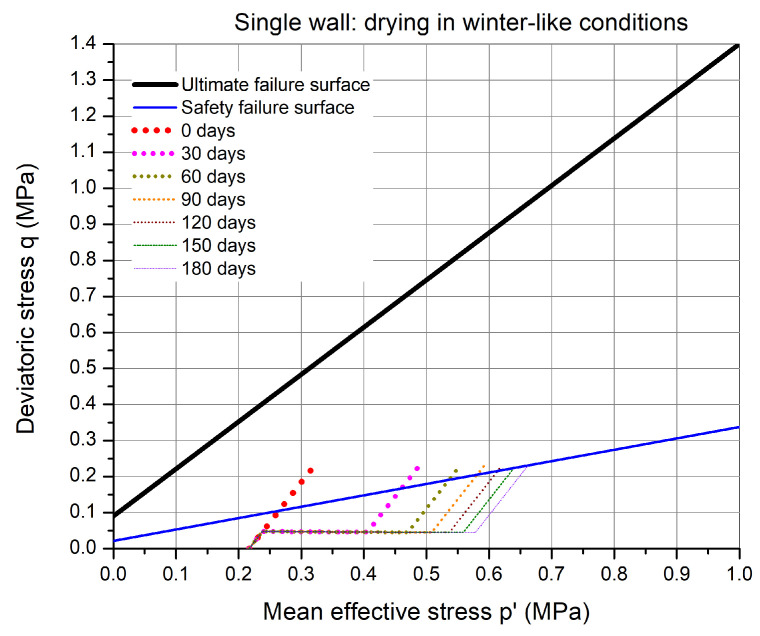
The stress path in $p^{\prime }$–*q* plane for the critical point in the wall at different duration of drying in winter-like environmental conditions.

**Figure 15 materials-15-00362-f015:**
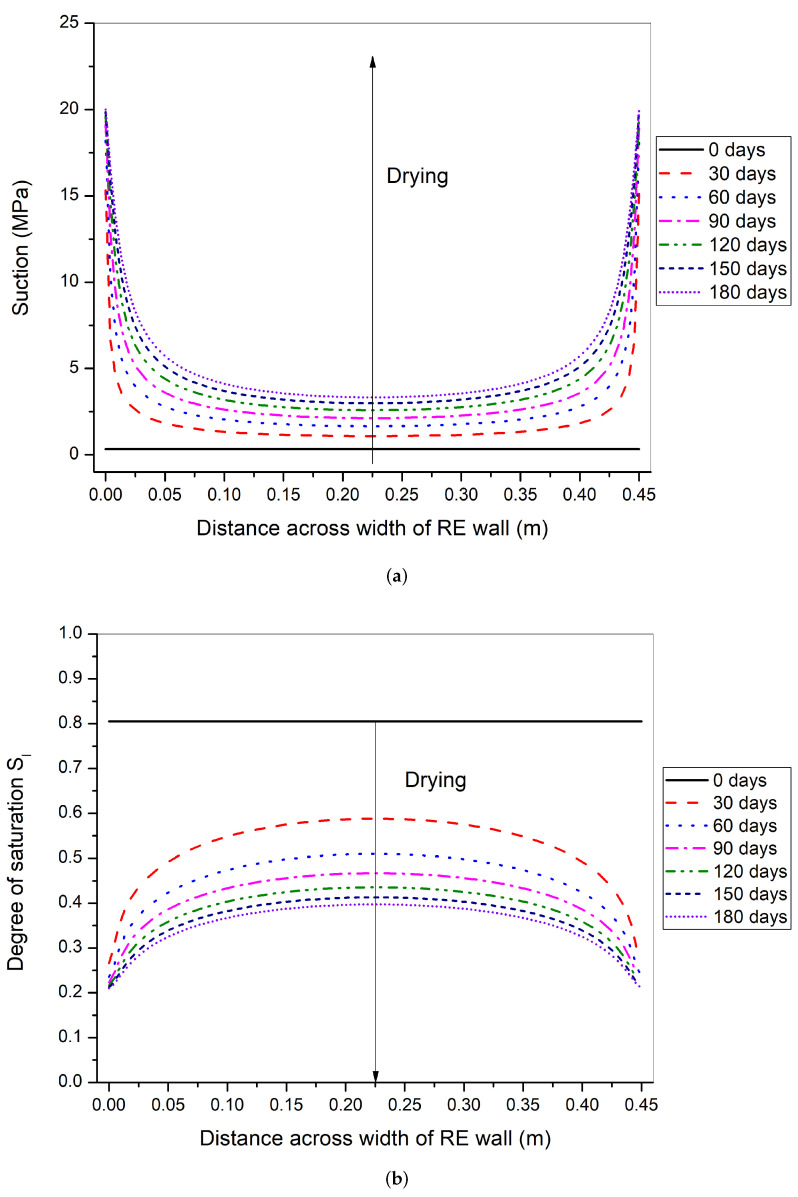
Suction (**a**) and degree of saturation (**b**) variation across the width of rammed earth wall at 0, 30, 60, 90, 120, 150, and 180 days of drying period in winter-like environmental conditions.

**Figure 16 materials-15-00362-f016:**
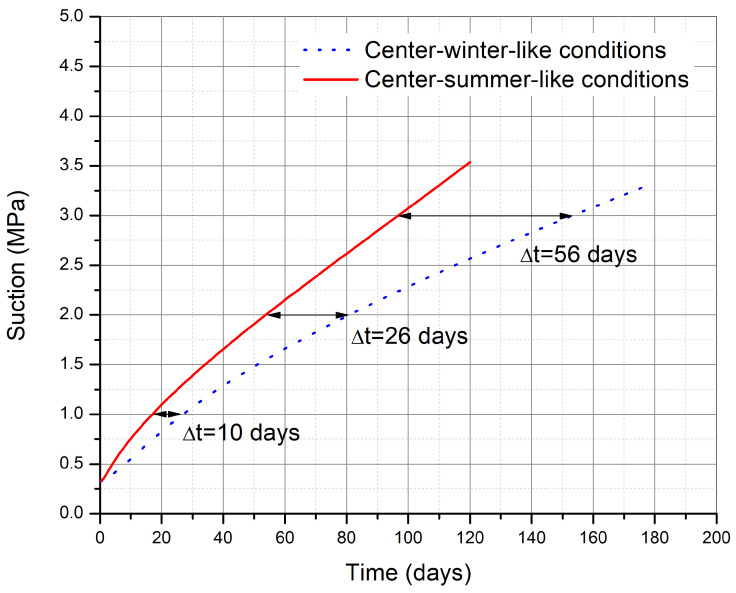
Comparison of suction evolution at the critical point (at the center) for summer-like and winter-like environmental conditions.

**Table 1 materials-15-00362-t001:** Different saline solutions, the relative humidity, and corresponding suction imposed at 25 °C.

Salt	KOH	${\mathrm{CH}}_{3}{\mathrm{CO}}_{2}K$	${\mathrm{MgCl}}_{2}$	NaBr	NaCl	KCl	$K_{2}{\mathrm{SO}}_{4}$
RH (%)	9	22.51	32.8	57.6	75.3	84.34	97.3
Suction (MPa)	331.3	205.3	153.4	75.9	39	23.4	3.8

**Table 2 materials-15-00362-t002:** Independent variables (unknowns) summary.

Variable Name	Notation	Equilibrium Equation of Resolution
displacements	u	Balance of momentum
liquid pressure	$P_{l}$	Mass balance of water
gas pressure	$P_{g}$	Mass balance of air
temperature	T	Internal energy balance

**Table 3 materials-15-00362-t003:** Various material parameters used for THM coupled problem and the method of determination of these parameters, where ‘Exp’ represents determined experimentally, ‘C’ represents that the classical or default value is chosen, and ‘L’ represents taken from the literature.

Parameters	Symbols	Units	Values	Determination
**Hydric and hygric parameters**				
VG retention curve parameter	$\alpha_{w}$	(MPa)${}^{-1}$	3.479	Exp
VG retention curve parameter	$n_{w}$	-	1.379	Exp
Intrinsic permeability	$k_{i}$	m${}^{2}$	5.7 × $10^{-16}$	Exp
Vapor diffusion parameter	*D*	m${}^{2}$ s${}^{-1}$ K${}^{-2.3}$ Pa	5.9 × $10^{-6}$	C [25]
Vapor diffusion parameter	*d*	-	2.3	C [25]
Tortuosity	$t_{s}$	-	1	C [25]
**Thermal parameters**				
Thermal conductivity of solid	$\lambda_{solid}$	(W m${}^{-1}$ K${}^{-1}$)	1.5	C
Thermal conductivity of liquid	$\lambda_{liq}$	(W m${}^{-1}$ K${}^{-1}$)	0.6	C
Thermal conductivity of gas	$\lambda_{gas}$	(W m${}^{-1}$ K${}^{-1}$)	0.025	L (GBP Pisé, 2018 [15])
**Mechanical parameters**				
Poisson’s ratio	ν	-	0.25	L [9,12]
Effective cohesion	$c^{\prime }$	kPa	43.9	Exp [16]
Effective friction angle	$\phi^{\prime }$	${}^{\circ }$	32.5	Exp [16]
Exponent of effective	α	-	1.9081	Exp [16]
stress parameter				
Porosity	$\phi_{p}$	-	0.291	Exp [26]

**Table 4 materials-15-00362-t004:** Synthesis of parameters in two different environemental conditions.

Env. Condition	T (°C)	Relative Humidity (%)	Suction (MPa)	$h_{m}$ (m/s)
summer-like	20	60	69.13	0.02
winter-like	5	85	20.86	0.02

## Data Availability

The data presented in this study are available on request from the corresponding author. The data are not publicly available due to the confidentiality order.

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
