# Peer review of "Benefit of Unsaturated Soil Mechanics Approach on the Modeling of Early-Age Behavior of Rammed Earth Building"

_materials, 2022, doi:10.3390/ma15010362_

Round 1

Reviewer 1 Report

Figure A1 needs translation to English; figure needs to be more precise, the protrusion length of the pitched roof is important;

Why does Figure 8 show the year 2019? Why is not a 3-5 year period taken instead?

Why is drying only determined in medium conditions, it is relevant during winter, during winter the worst edge condition should be used, not average temperatures

Reviewer 2 Report

The manuscript introduces a method to determine the drying period required for the walls to gain enough strength in order to build the subsequent floor. A coupled thermo-hydro-mechanical (THM) simulations are carried out on a single wall utilising the unsaturated soil mechanics approach and safety criterion. Simulated results showed that the safety criterion from the practical guide is very conservative and drying periods can be reduced without significantly compromising the safety factor.

I find the study is interesting and the results will be useful for readership. I provide my comments below for authors to consider for further improving the quality of the manuscript.

  • Abstract: Full description of ‘THM’ and ‘FEM’ should be mentioned at the first time.
  • Tables 1 and 3: It is not clear to me how those parameters were selected. Please include information or related references.
  • Equations for mass, momentum and energy balances are well-established in literature (Equations 2-8). Proper references or original sources are required.
  • A brief description of the CODE_BRIGHT with reference and original sources are suggested.
  • Figure 9: provide information/ justification why the critical point was selected at 0.5m from the base?
  • Figure 12: amend legend to ‘days’ instead of ‘D’ for a better description.
  • Figure A1: increase resolution of the Figure, some texts are not readable, and I suggest amending texts to English.

Round 2

Reviewer 2 Report

The authors have adequately attended comments and suggestion provided by the reviewer. I have now can recommended for the publication of the revised manuscript in Materials.

Author Response

Thank you for your valuable comments. A minor spell check has been done.

Regards

Parul Chauhan, Noémie Prime, Olivier Plé